

# Tectonic controls of Holocene erosion in a glaciated orogen

Byron A. Adams[1,2], Todd A. Ehlers[1]

[1]Department of Geosciences, Univsität Tübingen, D-72074, Germany
[2]Now at the School of Earth Sciences, University of Bristol, Bristol, BS8 1RJ, UK

*Correspondence to*: Byron A. Adams (byron.adams@bristol.ac.uk)

**Abstract.** Recent work has highlighted a strong, worldwide, glacial impact of orogen erosion rates over the last 2 Ma. While it may be assumed that glaciers increased erosion rates when active, the degree to which past glaciations influence Holocene erosion rates through the adjustment of topography is not known. In this study, we investigate the influence of long-term

tectonic and post-glacial topographic controls on erosion in a glaciated orogen, the Olympic Mountains, USA. We present 14 new [10]Be and [26]Al analyses which constrain Holocene erosion rates across the Olympic Mountains. Basin-averaged erosion rates scale with basin-averaged values of 5-km local relief, channel steepness, and hillslope angle throughout the range, similar to observations from non-glaciated orogens. These erosion rates are not related to mean annual precipitation or the marked change in Pleistocene alpine glacier size across the range, implying that glacier modification of topography and modern

precipitation parameters do not exert strong controls on these rates. Rather, we find that despite intense spatial variations in glacial modification of topography, patterns of recent erosion are similar to those from estimates of long-term tectonic rock uplift. This is consistent with a tectonic model where erosion and rock uplift patterns are controlled by the deformation of the Cascadia subduction zone.

**1 Introduction**

Before the onset of Late Cenozoic cooling and glaciation, the topographic expression of mountain belts resulted from tectonic processes, and the fluvial and hillslope processes which acted as the primary agents of erosion. High rock uplift rates in many of these ranges led to the buildup of topography and in some cases high relief, steep river channels and hillslopes, and commensurate high erosion rates (Willett, 1999). Because of the covariation between these characteristics, erosion rates

in fluvially dominated orogens have been shown to correlate with climatic and topographic metrics such as precipitation rate, relief, hillslope angle, and channel steepness via linear, non-linear and threshold relationships (Ahnert, 1970;Montgomery and Brandon, 2002;Ouimet et al., 2009;DiBiase et al., 2010). In the Late Cenozoic, alpine glaciers formed, and then advanced and receded many times due to climate oscillations. These glaciers possessed variable capacity to erode at the same rate as the rivers that existed before them, and regional rock uplift rates. In many mountain ranges, glaciers appear to have accelerated

erosion (Hallet et al., 1996;Shuster et al., 2005;Ehlers et al., 2006;Valla et al., 2011;Herman et al., 2013;Christeleit et al., 2017), while in other areas, glaciers may have done little to change erosion rates over the past few million years (Koppes and Montgomery, 2009;Thomson et al., 2010;Willenbring and von Blanckenburg, 2010).

As a result of Cenozoic climate change, the relationships between topographic metrics and observed Holocene (last ~12 kyr) erosion rates in glaciated mountain ranges are more complex than in purely fluvial settings (Moon et al., 2011;Godard

et al., 2012;Glotzbach et al., 2013). These poorly understood relationships are likely caused for two reasons. (1) Glaciers reorganized previously fluvial channel networks and relief to create a landscape with their preferred geometry, radically changing the orogen topography (Whipple et al., 1999;MacGregor et al., 2000;Brocklehurst and Whipple, 2002,





2004;Brocklehurst and Whipple, 2006;Anderson et al., 2006;Adams and Ehlers, 2017). (2) Holocene erosion rates may be dominated by transient signals as surface process remove the topographic disequilibrium imposed by glacial erosion (Moon

et al., 2011).

        In light of the previous studies, what remains uncertain is how much (if any) signal of tectonic processes can be discerned from a heavily glaciated orogen, and the degree to which common relationships between erosion and topographic metrics hold in post-glacial landscapes. Here we address this uncertainty and test the efficiency of Plio-Pleistocene glaciers to mask long-lived patterns of rock uplift as recorded by millennial-scale erosion rate estimates and modern topography. To

do so, we have conducted a systematic study of basin-averaged erosion rates from $^{26}$Al and $^{10}$Be concentrations in modern river sediments from the Olympic Mountains, USA (Fig. 1). The Olympic Mountains are well suited for this study because the efficiency of Plio-Pleistocene glaciers was controlled by spatially variable glacial mass balance, and the orogen has been shown to contain a wide range of rock uplift rates. We use our new data, in addition to $^{10}$Be concentrations from a previous study (Belmont et al., 2007), to investigate the spatial variations in erosion rates with respect to characteristics of the modern

topography including local relief, hillslope angle, and channel steepness, as well as precipitation. Further, we utilize $^{26}$Al/$^{10}$Be ratios, and new modelling efforts to investigate the degree to which cosmogenic nuclide inventories can accurately constrain erosion rates in glaciated mountain ranges.

## 2 Background

The Olympic Mountains are part of a chain of mountain ranges that define the forearc high of the Cascadia subduction zone (Fig. 1A). This forearc high marks the topographic and structural apex of an accretionary wedge which formed between the North American plate and the subducting Juan de Fuca plate (Tabor and Cady, 1978). The core of the range is comprised of an essentially unmetamorphosed, homogenous assemblage of medium- to fine-grained, greywacke interbedded with minor siltstone, mudstone, conglomerate, and basalt lenses. This group of rocks is referred to as the Olympic Subduction Complex,

and is located in the footwall of the Hurricane Ridge Fault (Fig. 1). Pillow basalts, breccias, volcaniclastic rocks and diabase make up the hanging wall of the fault, referred to as the Coast Range Terrane. Sedimentological and bedrock cooling histories suggest accelerated rock uplift and unroofing of the range began around 17-12 Ma (Tabor and Cady, 1978;Brandon et al., 1998). Rock uplift rates have been interpreted across the range from Neogene thermochronometric exhumation rates (apatite fission track) and Quaternary river incision rates (Pazzaglia and Brandon, 2001). These rates vary from ~300 m/Myr at the

fringes of the range to ~800 m/Myr in regions close to the geographic center of the range, forming a concentric pattern. Previous interpretations of these erosion rates suggest that they are governed by rock uplift rates (Brandon et al., 1998;Pazzaglia and Brandon, 2001;Batt et al., 2001). In most orogenic wedges, the rock velocity field is governed by the subducting plate geometry and convergence rate, as well as the pattern accreted materials from the subducting plate (Willett, 1999). The Olympic Mountains are thought to be no different (Batt et al., 2001); however, the subduction zone dynamics are

complicated by the significant arch in the subducting Juan de Fuca plate and the dome of accreted sedimentary units that make up the east-plunging Olympic Anticline (Brandon and Calderwood, 1990). Indeed, it is likely that these nuanced characteristics of the subduction zone may be responsible for the observed concentric pattern in erosion/rock uplift rates (Brandon et al., 1998;Pazzaglia and Brandon, 2001;Batt et al., 2001;Bendick and Ehlers, 2014).

        The Olympic Mountains have a general dome shape where the major drainages exhibit a radial pattern. The dome is

asymmetric where the locus of highest topography lies to the northeast of the range divide (Fig. 1B). Plio-Pleistocene alpine glaciers carved large valleys in the core of the range (Porter, 1964;Montgomery and Greenberg, 2000;Montgomery,



2002;Adams and Ehlers, 2017). The largest glaciers which occupied the Hoh, Queets and Quinault valleys all extended to the Pacific Ocean (Fig. 1B) (Thackray, 2001). Alpine glaciers were likely active in every valley of the Olympics; however, the size of the glaciers was highly variable (see glacial deposits Fig. 1B). Due to the W-SW prevailing wind direction and the

effects of topography on precipitation patterns, mean annual precipitation values decrease from ~6000 mm/yr in the southwest to less than 500 mm/yr in the northeast (Fig. 2A). This same precipitation gradient greatly influenced the Pleistocene equilibrium line altitude (ELA; the position where the ice flux in a glacier is at a maximum), and created an opposing pattern where the ELA increases at a rate of ~ 25 m/km toward the northeast (Porter, 1964) (Fig. 1B), thus controlling the size and efficiency of alpine glaciers (Adams and Ehlers, 2017). The range was bordered to the north and east by the Cordilleran Ice

Sheet (Fig. 1B) (Porter, 1964), which also likely restricted the size of alpine glaciers. While the impact of Plio-Pleistocene glaciation on more recent erosion has not been previously quantified, the suggestion of significant glacial erosion would imply that Holocene erosion rates may not simply be a function of rock velocities as suggested by older erosion histories discussed above.

## 3 Methods


### 3.1 Topographic analysis

Our topographic analysis is based on the 10 m National Elevation Dataset provided by the United States Geological Survey (www.ned.usgs.gov). Within this paper we calculate three topographic metrics which record relief at different spatial scales and controlled by different surface process – hillslope angle, local relief, and channel steepness. Each metric also has

strengths and weaknesses in quickly eroding, glaciated ranges. There is good evidence that hillslope angle values can reach maximum values due to the limitations of the internal angle of friction of hillslope materials. In such high erosion areas, hillslope angle values become insensitive to changes in erosion rates (Schmidt and Montgomery, 1995). Local relief values may also be limited in glaciated ranges due to the buzzsaw effect, whereby efficiently eroding glaciers increase the area near their ELA and thus control mean elevations and restrict relief locally (Meigs and Sauber, 2000;Brozović et al., 1997).

Our local relief ($R$) map (Fig. 2B) was made by calculating the difference between the highest and lowest elevations within a 5 km-diameter circular window. This form of relief is designed to encapsulate the relief of hillslopes and channels. The size of this window captures the elevation difference between peaks and valley floors of medium sized basins, but is small enough to detect changes in the relief structure of large drainage basins. The hillslope gradient ($S$) map (Fig. 2C) was calculated by finding the steepest angle of descent across a 3 x 3 pixel (30 x 30 m) square window.

The channel steepness map (Fig. 2C) was created by adjusting channel gradients ($S$) (m/m) by the non-linear change in downstream drainage area ($A$) (m$^2$) (Hack, 1957;Flint, 1974;Wobus et al., 2006):

$$S = k_s A^{-\theta} \qquad\qquad (1)$$

where $k_s$ is the channel steepness, and $\theta$ (dimensionless) is the channel concavity. This calculation normalizes slope values, for the concavity of the channel. For our calculations, we use $\theta = 0.45$. This value has been shown to describe the concavity

of fluvial systems in the Olympic Mountains (Adams and Ehlers, 2017). Since we utilize a single value of $\theta$, we report normalized channel steepness values ($k_{sn}$) (m$^{0.9}$). To report a mean value for a basin we calculated the mean normalized channel steepness for all portions of a basin, which are governed by fluvial processes. This generally occurs at drainage areas > 1 km$^2$.

Normalized channel steepness index ($k_{sn}$) analysis, which quantifies channel relief, has been used successfully in a

number of mixed fluvial and glacial landscapes as a fine scale measure of the erosion potential of glacial/fluvial processes in





a landscape (Montgomery, 2001;Brardinoni and Hassan, 2006, 2007;Brocklehurst and Whipple, 2007;Robl et al., 2008;Hobley et al., 2010;Glotzbach et al., 2013;Adams and Ehlers, 2017). Many assumptions generally that are adopted in purely fluvial settings do not apply in mixed glacial-fluvial landscapes. For instance, in our study we do not require that the Olympic Mountains are in topographic steady-state (where erosion and rock uplift at a point are balanced and therefore

elevations remain constant over time), nor do we imply that our slope-area analysis relates directly to the processes of glacial incision, or that rock uplift rates need to be spatially uniform. We emphasize that this technique provides a robust, geometric construct for understanding the importance of spatial changes in channel relief without demanding an understanding of all parameters within a specific incision law (fluvial or glacial). Unlike hillslope angle calculations, channel steepness values may be able to record changes in erosion/rock uplift rates in regions where hillslopes have reached a threshold (Ouimet et al.,

125 2009).

### 3.2 Processing sediment samples and calculating erosion rates

Basin-averaged erosion rates were calculated from concentrations of cosmogenic nuclides ($^{10}$Be and $^{26}$Al) in quartz sand from modern river basins throughout the Olympic Mountains (see Fig. S1 for sample location detail). This technique is

records the average erosion/denudation rate (we note that rates presented here incorporate both physical and chemical means of mass removal) integrated across the landscape upstream of the sample location (Brown et al., 1995;Granger et al., 1996;Bierman and Steig, 1996). Basins were selected to ensure a thorough sampling across precipitation, Pleistocene ELA, rock uplift, and topographic gradients. These basins are located within the Olympic Subduction Complex, where quartz is ubiquitous throughout the landscape.

Initial attempts to separate pure aliquots of quartz sand proved difficult due to the fine-grained nature of the lithologies found throughout the range. To reduce the need for aggressive hydrofluoric acid treatment, which would prematurely dissolve the quartz, we first disaggregated the 250-1000 μm sand fraction with a Selfrag, a high-voltage pulse disaggregater, at the University of Bern, Switzerland. From this stage on, samples were processed within the facilities at the University of Tübingen. After electronic disaggregation, sediments were re-sieved to 125-1000 μm and separated using a

strong magnetic field and then cleaned in concentrated, room temperature aqua regia for 24 hours. Samples were further cleaned in boiling pyrophosphoric acid and then boiling sodium hydroxide at least 3 times. The quartz was then leached 1% hydrofluoric acid while in an ultrasonic bath for 1 week. A final leach was performed on the samples with concentrated hydrofluoric acid before spiking with beryllium. Samples were not spiked with aluminum. Beryllium and aluminum were separated, oxidized and loaded into cathodes for mass spectrometer analysis using established protocols (Von Blanckenburg

et al., 1996). Native Al concentrations within samples were measured with an inductively-coupled plasma optical emission spectrometer at the University of Tübingen. Beryllium and aluminum ratios were measured at the University of Cologne Centre for Accelerator Mass Spectrometry.

To calculate erosion rates, we followed the approach of Portenga and Bierman (2011), which simplifies each basin to a single point where the production rate is equal to the mean production rate of the entire basin, enabling the use of the

CRONUS online calculator (Balco et al., 2008). Basin-averaged production rates were based on the elevation and latitude of each pixel in a basin using the scheme of Stone (2000). The effective elevation and latitude of each basin are the elevation and latitude values corresponding to this mean scaling factor (Table S1). We calculated topographic shielding due to obstacles according to the equations of Dunne et al. (Dunne et al., 1999), and snow shielding from the equations of Gosse and Phillips (Gosse and Phillips, 2001). Pixels under extant ice are assumed to be 100% shielded. Our snow depth maps are based on



satellite snow cover data that were calibrated by snow depth observations in the Olympic Mountains (see Supplement for more details). For CRONUS calculations, the following inputs were used: Elevation Flag = std, Thickness = 1 cm, Density = 2.7 g/cm$^3$, Be Standard = 07KNSTD, Al Standard = KNSTD.

Because our methods to calculate effective latitude and elevation do not incorporate temporal production fluctuations, we report erosion rates from the CRONUS calculator from the constant production rates determined by Lal
(1991) and Stone (2000). To enable comparison between new and previous measurements, we recalculated erosion rates from 7 sand samples within the Olympic Mountains previously reported by Belmont et al. (2007).

**3.3 Isotopic equilibrium modeling**

The application of detrital cosmogenic nuclide techniques as an estimator of basin-averaged erosion rates in post-
glacial landscapes is not yet a common practice as there is a high potential for violation of the assumptions inherit to the calculation of erosion rates from nuclide concentrations. The assumptions that can be most problematic for a glacial terrain are: the eroding materials are in isotopic equilibrium; and modern river sediment is spatially and temporally representative of all sediment in the basin. To explore the nature of isotopic equilibrium we describe new modeling efforts in the this section. The topic of sediment storage a mixing is discussed in Sections 4.2 and 5.1. Surface materials are in isotopic equilibrium
when the production of cosmogenic nuclides is balanced by their removal through erosion and radioactive decay, in this state the concentration of nuclides in surface materials is steady over time. Since glacial ice intercepts, and thus shields underlying material from cosmic radiation, previously and currently glaciated basins may violate the isotopic equilibrium assumption if ice was present recently, and erosion has not been able to remove the older shielding signal (Moon et al., 2011;Portenga et al., 2014;Vance et al., 2003). Therefore, interpreting cosmogenic nuclide concentrations as direct measurements of erosion in
some glaciated landscapes can lead to overestimated rates (Gosse and Phillips, 2001;Vance et al., 2003;Portenga et al., 2014).

To test the hypothesis that our samples are in isotopic equilibrium we conducted a suite of numerical models to constrain the evolution of the concentration of cosmogenically produced nuclides at depth, starting at a time just after a period when ice completely shielded surface production. When the model starts, production occurs according to the equations of (Anderson et al., 1996):

$$N(z,t) = N_0 e^{-\lambda t} + \frac{P_0 e^{-\rho z/\Lambda}}{\lambda + \rho E/\Lambda}(1 - e^{-(\lambda + \rho E/\Lambda)t})$$

180                                                                                                                    (2)

where $N$ is the cosmogenic nuclide concentration (atoms/g), $t$ is time (yr), $N_0$ is the inherited concentration cosmogenic nuclides (atoms/g), $E$ is the erosion rate (cm/yr), $\lambda$ is the decay constant for [10]Be (1/yr), $z$ is the depth below the surface (cm), $P_0$ is the [10]Be production rate at the surface (atoms/g/yr), $\Lambda$ is the attenuation length for cosmogenic nuclide production (g/cm), and $\rho$ is the material density (g/cm$^3$). In our models the following values were used: $\lambda$ = 4.99e-7 1/yr, $P_0$ = 10 atoms/g/yr
(though the results of this model are not sensitive to this value), $\Lambda$ = 160 g/cm, and $\rho$ = 2.7 g/cm$^3$.

Using a finite difference method, the model runs forward from the time of since unshielding, and surface concentrations increase over time as production occurs, and deeply shielded materials are eroded from the top of the model. The concentration at the surface is compared to the steady-state value to assess the approach toward isotopic equilibrium. A range of erosion rates which span the observed erosion rates in this study are tested.


**3.4 Relationships between erosion rates and basin parameters**



We performed non-linear regressions on our new and existing erosion rate data. To provide a better sense of the distribution of topographic metrics within a basin, we provide box-and-whisker plots within our bivariate plots, though our regressions discussed in the following sections are based on mean statistics. During our regression analysis, we chose to
process all data together, as opposed to subsampling populations. This was decided to avoid the issue of small sample statistical bias, and because it was not obvious how to subsample our data *a priori*, without adding bias. These regressions included the influence of uncertainties on both variables. Goodness of fit values were determined by the mean square weighted deviation (MSWD). For well-fit data, MSWD values tend toward 1 within an uncertainty based on the degrees of freedom (based primarily on the number of samples). Elevated MSWD values, are caused by the high degree of inter-sample variability,
and suggest (1) the two variables shown in the plot are not highly correlated, (2) a more complex function exists between the two variables, or (3) that uncertainties are underestimated.

As a means to assess the relationship between erosion and hillslope processes we use a variation of the non-linear relationship proposed by Roering et al. (2001), which captures the effects of diffusive processes and landsliding:

$$E = \frac{KS}{1-\left(S/S_c\right)^2} \tag{3}$$

where $E$ is the basin-averaged erosion rate (m/Myr), $K$ is a rate constant related to the diffusivity of the eroding material (m/Myr), $S$ is the basin-averaged hillslope gradient (m/m), and $S_c$ is a critical slope at which soil flux approaches infinity (m/m).

We used a similar equation from Montgomery and Brandon (2002) to explore the relationship between erosion rates and 5-km local relief:

$$E = \frac{KR}{1-\left(R/\ R_c\right)^2} \tag{4}$$

where $K$ is a different rate constant (m/Myr), $R$ is the basin-averaged local relief normalized by the diameter of the moving window (m/m), and $R_c$ is a limit to the possible values of local relief normalized by the diameter of the moving window (m/m).

Previous studies have suggested that that channel steepness values can vary spatially according to a relationship with basin-averaged erosion rates through a stochastic threshold model of fluvial channel incision (DiBiase et al., 2010). Such a
model generally produces a non-linear relationship. However, using a model based solely on fluvial incision in the Olympic Mountains would be misleading as the modern river channel likely still reflect the preferred geometry of Plio-Pleistocene glaciers (Adams and Ehlers, 2017). Instead, we implemented a least-squares power function regression to explore possible connections between erosion and channel steepness, similar to other recent studies (Scherler et al., 2013):

$$E = C k_{sn}{}^{p} \tag{5}$$

where $C$ is the pre-exponential coefficient, $k_{sn}$ is the mean channel steepness value of the basin, and $p$ is the power. We used the same least-squares routine to analyze the relationship between erosion and precipitation (e.g. replace $k_{sn}$ with the mean precipitation in Eq. 5).

## 4 Results

### 4.1 Topographic analysis

The topography of the Olympic Mountains is a mixture of high glacial cirque basins, wide and flat-floored valleys at low elevations, and very steep landscapes in between. This juxtaposition of varied landscapes creates skewed and multi-modal distributions of topographic metrics within drainage basins throughout the range (Table 1 and 2, Fig. 2B, 2C and 2D).




While it is useful to report arithmetic means of basin statistics to simplify a landscape, it can often be difficult to constrain the

significance of such means in the context of their relation to erosion rates. In complex landscapes not defined by uniform and

steady surface processes, like the Olympic Mountains, normally distributed topographic metrics with good central tendency

are unlikely, especially for metrics which capture fine spatial scale processes like those occurring at the scale of hillslopes

and channel segments. To provide a better sense of these distributions, we have included simplified histograms next to our

reported statistics in Table 1 and 2. Because of this size limitation we are not able to calculate an accurate channel steepness

value for one of basins from a previous study, U-WC (Belmont et al., 2007).

Basin-averaged hillslope angles are generally high, in most cases above 28°, as is the standard deviation of hillslope

angles within each basin (mean $2\sigma = 21°$) (Table 2 and Fig. 2C). Basin-averaged channel steepness values range between 23

and 181 m$^{0.9}$, and also have proportionally large standard deviations (mean $2\sigma = 89$ m$^{0.9}$) (Table 2 and Fig. 2D). Basin-

averaged local relief values (calculated within a 5-km diameter window) range between 350 and 1443 m (Table 2 and Fig.

2B). Relative to hillslope angle and channel steepness values, local relief values exhibit smaller variance within sampled

basins (mean $2\sigma = 219$ m). The lower-elevation basins on the western flank of the range, which evaded Last Glacial Maximum

(LGM) alpine glaciers (Thackray, 2001;Belmont et al., 2007), have the lowest topographic metric values of the sampled basins

(8 basins: mean $R = 544$, mean $S = 23$, mean $k_{sn} = 43$). The mean values from the 8 glaciated, west side basins and the 6

glaciated, east side basins are effectively the same: mean $R = 1296$, mean $S = 31$, mean $k_{sn} = 151$; and mean $R = 1239$, mean

$S = 31$, and mean $k_{sn} = 143$, respectively. Despite the rain shadow and the significant discrepancy in the size of alpine glaciers

across the range divide, there is no difference in topographic metrics within the rugged core of the range across the divide.

There is a high degree of correlation between some basin-averaged precipitation values and basin-averaged elevation

(Fig. 3A.), as would be expected from the PRSIM precipitation dataset which includes an orographic precipitation model to

do reanalysis simulations (Daly et al., 1994). This correlation is only strong on the western flank of the mountain where the

topographic and precipitation gradients are smoothest. These same sub-group of basins also exhibit a strong correlation

between basin-averaged hillslope angle and basin-averaged elevation (Fig. 3B). However, there is no correlation between

elevation and precipitation or hillslope angle in the core of the range. There is good correlation between basin-averaged

elevation and both basin-averaged local relief and channel steepness (Fig. 3C-D), across the range.

## 4.2 Cosmogenic basin-averaged erosion rates


While we present both [10]Be and [26]Al data (Table 3, see Tables S1 and S2 for complete nuclide analysis), we focus

our analysis on erosion rates derived from [10]Be in this study (Fig. 2E) to provide a means of comparison to existing data from

the Olympic Mountains (Belmont et al., 2007). To a first order, basins located at elevations < 1000 m have been eroding at

slow rates, all less than 240 m/Myr, whereas basins in the higher, rugged core of the range have higher erosion rates reaching

over 1400 m/Myr (Table 3 and Fig. 2E). We obtained the highest apparent erosion rates (> 1500 m/Myr) from the flanks of

Mount Olympus, whose drainages contain the largest extant glaciers in the Olympic Mountains (basins WA1519, WA1523,

WA1524). However, the very low [10]Be concentrations (i.e. high apparent erosion rates) from Mount Olympus may be a

signature of isotopic disequilibrium. Samples WA1519, WA1523 and WA1524 come from basins which likely contained

thick ice the longest, and still have small valley glaciers today. The [10]Be abundances for these three basins only range from

5-7 times the [10]Be blanks. These low abundances are likely caused by the shielding of rock and soil below glaciers. Such low

measurements not only increase in the internal uncertainty of the concentration calculation, but also raises questions about

the accuracy of the erosion rate calculation and interpretation. For this reason, we do not include these basins in our regression





analysis.

Sample $^{26}$Al/$^{10}$Be ratios from the Olympic Mountains mostly vary between 8.5 ± 3.5 (2σ) and 4.7 ± 1.6 (2σ) (Table
3 and Fig. 4). Nearly all samples have $^{26}$Al/$^{10}$Be ratios that are statistically indistinguishable from the expected naturally
occurring ratio (6.75) (Table 3 and Fig. 4), suggesting that the sediments in our samples record a relatively simple erosion
rate history over the integration time. As such, there is no significant influence of reworking older sediments in our
measurements. Furthermore, because our erosion rate calculations assume a natural production rate ratio of 6.75, and our
measured ratios are mostly indistinguishable from this value, $^{10}$Be and $^{26}$Al derived erosion rates are statistically
indistinguishable, though the $^{26}$Al derived rates are much less precise (Table 3). Two samples from Mount Olympus basins,
WA1519 and WA1523, have much lower ratios.

Snow shielding can reduce production rates, and therefore, reduce calculated erosion rates by up to 16% in the core
of the range, but only ~3% reduction is found in lower elevation basins on the western flank (Table S3). While it is difficult
to assess our snow shielding estimates, we note the relative effect on erosion rates is similar to those based on snow-depth
measurements within other snowy orogens (Wittmann et al., 2007;Norton et al., 2010;Scherler et al., 2013).

### 4.3 Isotopic equilibrium modeling

As seen in Eq. 2 the likelihood of being in isotopic equilibrium for any cosmogenic radionuclide is mostly controlled
by the time since deglaciation and the local erosion rate (assuming an inheritance of zero). Figure 5 illustrates that quickly
eroding terrains more quickly remove ice-shielded materials, and thus, these terrains can each a new equilibrium state faster
after the ice recedes. In fact, our model output suggests that at relatively low erosion rates (~100 m/Myr), terrains can achieve
isotopic equilibrium in a few thousand years. These results suggest that the cosmogenic nuclide inventories from many
glaciated landscapes on Earth could record accurate erosion rates (barring other complicating factors).

### 4.4 Relationships between erosion rates and basin parameters

Our best-fit curve (MSWD = 17) suggests the observed relationship between hillslope gradient and erosion is
controlled by a critical slope value of 37° and a rate constant of 250 m/Myr (Fig. 6B). These parameters fit our data
considerably better than the previous boundary conditions suggested by Montgomery and Brandon (2002) for the Olympic
Mountains ($S_c$ = 40°, $K$ = 500 m/Myr, MSWD = 54). Our regressions also record a limited local relief of 1820 m ($K$ = 0.24
m/Myr, MSWD = 4.3) (Fig. 5A). These parameters are also different than those of Montgomery and Brandon (2002) based
on rates from low-temperature thermochronometry ($R_c$ = 1500, $K$ = 0.25 m/Myr, MSWD = 13). Regressions from the least-
squares technique shows a best-fit, nearly-linear model (i.e. the exponent is 0.98) for the relationship between erosion and
channel steepness (Fig. 5C). The least-squares technique demonstrates that there is no strong linear or non-linear relationship
between erosion and precipitation across the range (Fig. 5D).


### 5 Discussion

### 5.1 Reliability of cosmogenic erosion rates in the glaciated Olympic Mountains

While there are still a few minor valley/cirque glaciers, the Last Glacial Maximum occurred in the Olympics
Mountains ~17 ka (Thackray, 2008). Therefore, most of our samples should largely reflect post-recessional erosion rates. Our
model results show that even the slowest eroding landscapes in the Olympic Mountains could achieve isotopic equilibrium
within ~3000 yrs (Fig. 5). Furthermore, the slowest eroding basins from the western flank of the range did not contain valley



glaciers during the LGM (Thackray, 2001), and thus, these samples are even less likely to violate this assumption. The most recently deglaciated portions of the range are in the rugged core, where erosion rates are also higher, and where some landscapes can reach isotopic equilibrium in less than 1500 yrs (Fig. 5).

In landscapes where the cosmogenic nuclide inventories are a function of constant exposure or constant erosion, the ratio of $^{26}$Al to $^{10}$Be within sediments can be predicted based on the modeled (Lal, 1991;Balco et al., 2008) or measured (Corbett et al., 2017). Most recently studies suggest that our samples should have natural $^{26}$Al/$^{10}$Be ratios of ~6.75 (Balco et al., 2008). Based on our measured $^{26}$Al/$^{10}$Be ratios (Figure 4), we find it unlikely that sediment storage and reworking (e.g. from terraces or moraines) has violated our assumptions that modern sediments record a representative sample of all sediment

in the basin. If anomalously low concentration quartz was introduced into our samples via incision of older deposits (glacial or fluvial), or through deep landslides, we would expect to see depressed $^{26}$Al/$^{10}$Be ratios.

### 5.2 Interpreting relationships between erosion and basin metrics

     In landscapes with high fluvial and/or glacial erosion, soil production and hillslope transport may not be able to

adjust to channel incision. In such a case, hillslope angles steepen and tend toward a threshold that is controlled by the strength of the material (Schmidt and Montgomery, 1995). Once hillslopes reach such a threshold, increases in erosion can only occur with a commensurate increase in hillslope failure (Burbank et al., 1996), and the form of these hillslopes are no longer sensitive to changes in erosion. However, the gradients of channels in these very steep landscapes are generally much lower than the internal angel of friction, and as such, still have the capacity to adjust to increases in erosion rate. Therefore, it has been

suggested that the morphology of channels is a more robust metric to detect erosion rate variations (Ouimet et al., 2009;DiBiase et al., 2010).

     Our data show that basin-averaged hillslope gradients cease to increase in basins eroding faster than ~300 m/Myr Fig. 6B). This limit has been observed in many other landscapes around the world (DiBiase et al., 2010;Ouimet et al., 2009;Montgomery and Brandon, 2002;Binnie et al., 2007). Basin-averaged hillslope angle values tend to reach a maximum

around 34°, as also shown by Montgomery (2001) using 100 km$^2$ grids across the Olympic range. The extent to which these threshold hillslope angles are indicative of rock uplift rates or glacial modification is not completely clear. While it is possible that the weak lithologies and fast erosion rates of the Olympic Mountains may be setting these threshold hillslopes, it has also been documented that hillslope angles have likely been increased throughout the range via glaciers widening valleys (Montgomery, 2002;Adams and Ehlers, 2017), or eroding headward and migrating ridge tops (Adams and Ehlers, 2017).

Similarly, basin-averaged local relief values do not exceed ~1350 m despite increasing erosion rates (Fig. 6A). This apparent threshold relief may be due to the influence of the glacial buzzsaw effect, whereby efficiently eroding alpine glaciers have controlled the mean and range of elevations during the Plio-Pleistocene. If these local relief values are limited due to glacial incision, then this would be a transient topographic signal, and imply that local relief could have been higher in the past. As such, we do not suggest that the non-linear fit parameters for hillslope and local relief data presented here are related

to topographic steady-state conditions; however, our fit parameters are not very different from those relating topography to long-term erosion rates (Montgomery and Brandon, 2002). Glaciers may have also reduced channel steepness values while active in the Plio-Pleistocene by incising into channel floors more deeply than rivers had previously (Adams and Ehlers, 2017). This effect may be seen in the apparent limit of channel steepness around 160 m$^{0.9}$ (Figure 6C).

     What is clear from these regressions is that in as much as relationships between modern topography and erosion exist

based on thermochronometric data in the Olympic Mountains (Montgomery and Brandon, 2002), so do relationships between



modern topography and detrital cosmogenic erosion rates. One advantage to using detrital studies is the obvious choice for an erosion integration area (i.e. the average erosion rate is integrated across the area of the sampled basin), as opposed to selecting a given area around a specific point in the landscape for a bedrock sample. Indeed, subtle changes in the sampling area throughout the Olympic Mountains can have a large influence on the calculated topographic metric (i.e. changing the

radius of a circle around a point can add topography across a drainage divide). However, there is a greater uncertainty regarding the integration timescale of cosmogenic rates in that it can often only be assumed that rates only integrate over hundreds to thousands of years. Our analyses provide good evidence for relationships between topographic metrics and basin-average erosion rates, which are likely the result of long-lived Miocene tectonics (Brandon et al., 1998), and Plio-Pleistocene climate change (e.g. hillslope gradient, local relief, channel steepness) (Porter, 1964;Montgomery and Greenberg,

2000;Montgomery, 2002;Adams and Ehlers, 2017). The key question remaining for this study area, and similarly glaciated and tectonically active orogens elsewhere is - what are the controls on post-glacial erosion rates?

**5.2 Orogenic processes governing erosion rates**

With our new and previously published erosion rates, we have made several important observations in the previous
sections that we elaborate on below. These observations are: 1) There is no relationship between precipitation and Holocene erosion rates across the range (Fig. 6D). 2) Basins with similar topographic characteristics have equivalent erosion rates, even across the range divide where glacial size changed drastically (compare black and grey samples in Fig. 6). 3) It is apparent from our regressions there are non-linear relationships with local relief, channel steepness and hillslope angle, and Holocene erosion rates (Fig. 6). In tectonically active and previously glaciated mountain ranges there are three common orogenic
processes that are most often suggested to dominate Holocene erosion rate patterns: climate gradients (Carretier et al., 2013;Olen et al., 2016), glacial modification of the landscape (Moon et al., 2011;Glotzbach et al., 2013), and patterns of tectonic rock uplift (Adams et al., 2016;Scherler et al., 2013;Godard et al., 2014). In the following we explore the relevance and applicability of these explanations to our data set.

First, we find it highly unlikely that a precipitation gradient similar to the modern has a significant control on recent
erosion rates. There is no clear relationship between modern precipitation and erosion rates (Fig. 6D). Even in the neighboring, glaciated Cascade Range (~70 km to the east of the Olympic Mountains) where the modern precipitation gradient is not as large, there is a strong linear relationship suggesting erosion scales with precipitation over diverse timescales, thus making it an important condition for setting Holocene and older erosion rates (Moon et al., 2011;Reiners et al., 2003).

Second, our data do not suggest that destabilization of the landscape via glacial incision has played a primary role in
setting the Holocene erosion pattern. Despite the significant gradients in the Pleistocene ELA (Fig. 2E) and the change in glacier size across the range divide (Fig. 1B), there is no statistical difference between the erosion rates across the range divide for basins of similar topographic characteristics (Fig. 2 and Fig. 5A, 5B, 5C), and there is a very weak correlation between ELA and erosion rates (Fig. S2). However, it has been observed that Plio-Pleistocene glaciers widened valleys in the Olympic Mountains (Montgomery, 2002;Adams and Ehlers, 2017), which led to the lengthening and steepening of hillslopes
throughout the range. In the nearby Cascade Range, similar valley widening has led to hillslopes with higher likelihoods for failure (Moon et al., 2011). Unlike in the Olympic Mountains, findings from the Cascades suggest that the range was heavily influenced by glacial incision to an extent that the topographic form largely reflects relict glacial processes, and as a result, Holocene erosion rates are more likely to be correlated with precipitation in these landscapes further from equilibrium. Our analysis suggest that the changes in the landscape due to Plio-Pleistocene glaciation in the Olympic Mountains likely only





steepened relatively small areas of hillslopes of landscapes relative to the already steep conditions imposed by high rock uplift and erosion rates. Similarly, glacial incision may have only influenced small relatively small portions of the channel network and range relief, which might appear as threshold values of channel steepness and local relief, or simply to make the distributions of these parameters within a basin more complex. Therefore, the landscapes examined the Olympics may have been only moderately perturbed by Plio-Pleistocene glacial incision, and they may still record a relatively close balance

between recent erosion rates and rock uplift.

Taken together, we suggest that the Holocene erosion rates (Fig. 2E), mean elevation, local relief (Fig. 2B), and channel steepness (Fig. 2D) observed in the Olympic Mountains most closely record a rock uplift pattern that increases from the low-relief flanks to the rugged core of the range (Fig. 7), similar to what has been shown in other datasets (Brandon et al., 1998;Batt et al., 2001;Pazzaglia and Brandon, 2001). Figure 7 illustrates the similarity of the trends in other rock uplift rate

proxies, and the cosmogenic erosion rates presented here in a direction parallel to the convergence across the Olympic Mountain range. Adams and Ehlers (2017) proposed that a spatial pattern of rock uplift similar to the one described above was consistent with the observations of the bend in the subducting Juan de Fuca plate at the Cascadia subduction zone and the dome of accreted sediments in the core of the Olympic Mountains, which form the east-plunging Olympic Anticline (Brandon and Calderwood, 1990). This pattern of focused rock uplift and erosion is also predicted for the geometry of the

curved subducting Juan de Fuca plate (Crosson and Owens, 1987;Bendick and Ehlers, 2014).  Our observations are in line with previous authors who have highlighted importance of subduction zone dynamics for setting the pace and pattern of erosion in the Olympic Mountains (Brandon and Vance, 1992;Pazzaglia and Brandon, 2001;Batt et al., 2001;Brandon et al., 1998;Stolar et al., 2007).

**6 Conclusions**

Whether post-glacial erosion rates balance longer-lived rock uplift rates is strongly influenced by whether post-glacial climate changes (e.g. increase or decrease in precipitation) or topographic perturbations (e.g. hillslope steeping or channel shallowing) have changed the activity of extant surface processes as compared to before the onset of glaciation. There are many examples of ranges where there have been significant changes to topography [*Adams and Ehlers*, 2017; *Brardinoni*

*and Hassan*, 2006; 2007; *Simon H Brocklehurst and Whipple*, 2007; *Glotzbach et al.*, 2013; *Hobley et al.*, 2010; *D. R. Montgomery*, 2001; *Robl et al.*, 2008] and erosion during and after glaciation (Moon et al., 2011;Reiners et al., 2003;Christeleit et al., 2017), and others where such changes are not clearly observed (Thomson et al., 2010). More generally, these changes were explored in a coupled ice dynamic/landscape evolution model testing the modification of topography and erosion rates due to alpine glaciation (Yanites and Ehlers, 2012). The results of this study suggest that the degree of erosion change before

and after glaciation is a function of regional temperature and the rock uplift rate. These two parameters control the glaciers ability concentrate elevations at or near the ELA were ice erodes efficiently. If too much or too little of the landscape lies above the ELA, then glacial erosion is not very efficient and little topographic perturbation occurs. In these landscapes, erosion rates may change during glacial periods, but interglacial erosion rates return to near rock uplift rates, as before. In the cases where glaciers were very effective agents of erosion, relief (on hillslopes or channels) is reduced during glaciation, and post-

glacial erosion rates can be lower than pre-glacial, and vice versa. As such, there is a sweet spot within mountain range conditions where glaciers are more efficient, and furthermore, even when glaciers are efficient it cannot be assumed how post-glacial rates might change. To put it another way, it should be assumed that landscapes will respond differently alpine glaciation depending on climate and topographic conditions before and after glaciation.



When our new Holocene erosion rate pattern is compared with older patterns of estimated rock uplift rates (Fig. 7),
there are a few apparent mismatches. In some locations, our rates are higher and lower than rock uplift rates (as might be
expected in post-glacial landscapes), but overall the pattern of increasing rates from the flanks to the core of the range is
consistent between these datasets. We suggest this long-lasting pattern is primarily controlled by tectonic forces, and while
the Plio-Pleistocene alpine glaciers of the Olympic Mountains have created a wonderfully sculpted landscape, they have not
radically altered the topography enough to drastically change the pattern of erosion.


**Acknowledgments**

We thank Lorenz Michel, Holger Sprengel, Roger Hoffman, Bill Baccus, Jerry Freilich and the Olympic National Park
Rangers for assistance while in the field and logistics. We also acknowledge the help of Christine Lempe, Hella Whitmann,
Mirjam Schaller, Dagmar Kost, and Jessica Starke with the processing of these stubborn samples. We grateful to Karl Lang,
Matthew Jungers, and Mirjam Schaller for fruitful discussions. Frank Pazzaglia is thanked for his comments on an earlier
version of this manuscript. This work was supported by a European Research Council (ERC) Consolidator Grant number
615703 to T. Ehlers. All data used in the manuscript are freely available in either the manuscript tables or online supplemental
material. The authors declare that they have no conflict of interest.

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





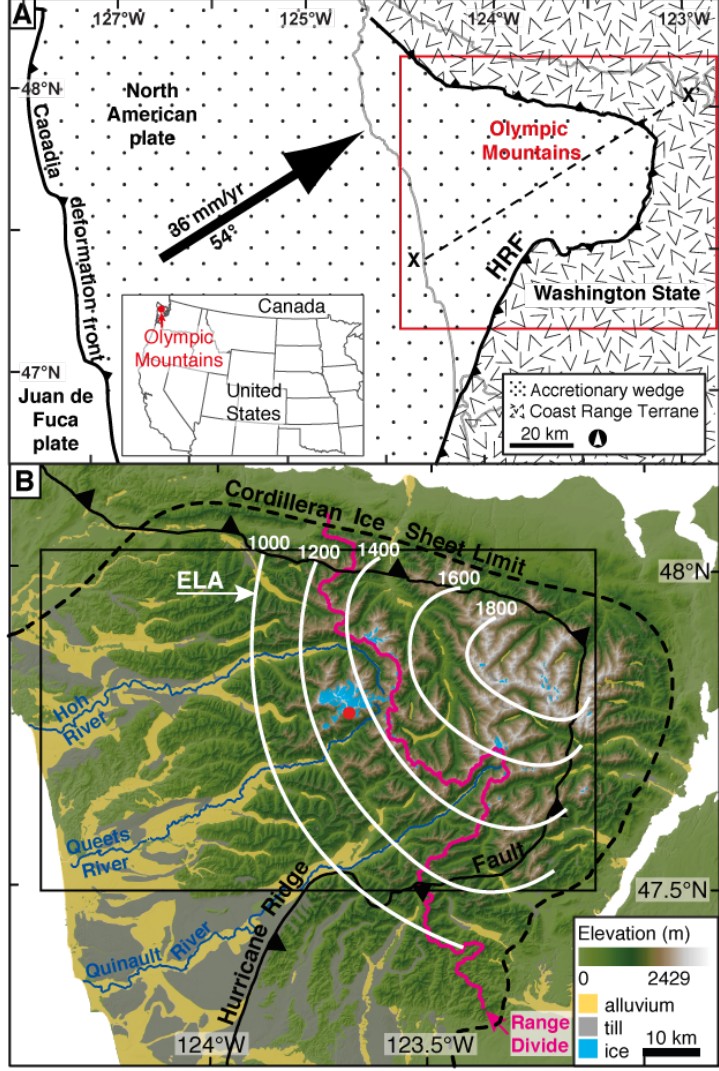

**Figure 1. Topographic and geologic features of the Olympic Peninsula, Washington State, USA.** A) Simplified geology based on Brandon et al. (1998). The relative velocity of the Juan de Fuca plate toward the North American plate is ~36 mm/yr with a bearing of ~54° (DeMets and Dixon, 1999). Red box denotes the extent of panel B. HRF – Hurricane Ridge Fault. Grey lines outline the coast of Washington State. Dashed black line is the cross-section line for Fig. 7. B) Elevation map of the Olympic Mountains. Ice, including extant alpine glaciers, is masked in blue. Undifferentiated Quaternary alpine glacial till and alluvial deposits are marked in grey and yellow, respectively. Contours of equilibrium line altitudes (ELA) from Porter (1964) are denoted by white lines (values shown in meters above sea level). Black box denotes the extent of Fig. 2 panels. A red dot marks Mount Olympus.



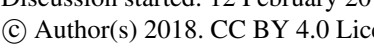

**Figure 2. Attribute and erosion maps of the Olympic Mountains.** Solid outlines denote the boundaries of sampled basins. High rugged core outlined in black/white dashed lines. A) Mean annual precipitation (MAP) map based on PRISM data (Daly et al., 1994). Open and closed circles mark new and previously published sample locations, respectively. B) Local relief (5-km relief) map. C) Hillslope angle map. D) Channel steepness ($k_{sn}$) map plotted for accumulation areas > 2 km$^2$. E) Basin-averaged erosion rate map. Range divide marked in magenta. Equilibrium line altitude (ELA) contours from Porter (Porter, 1964) are in black.



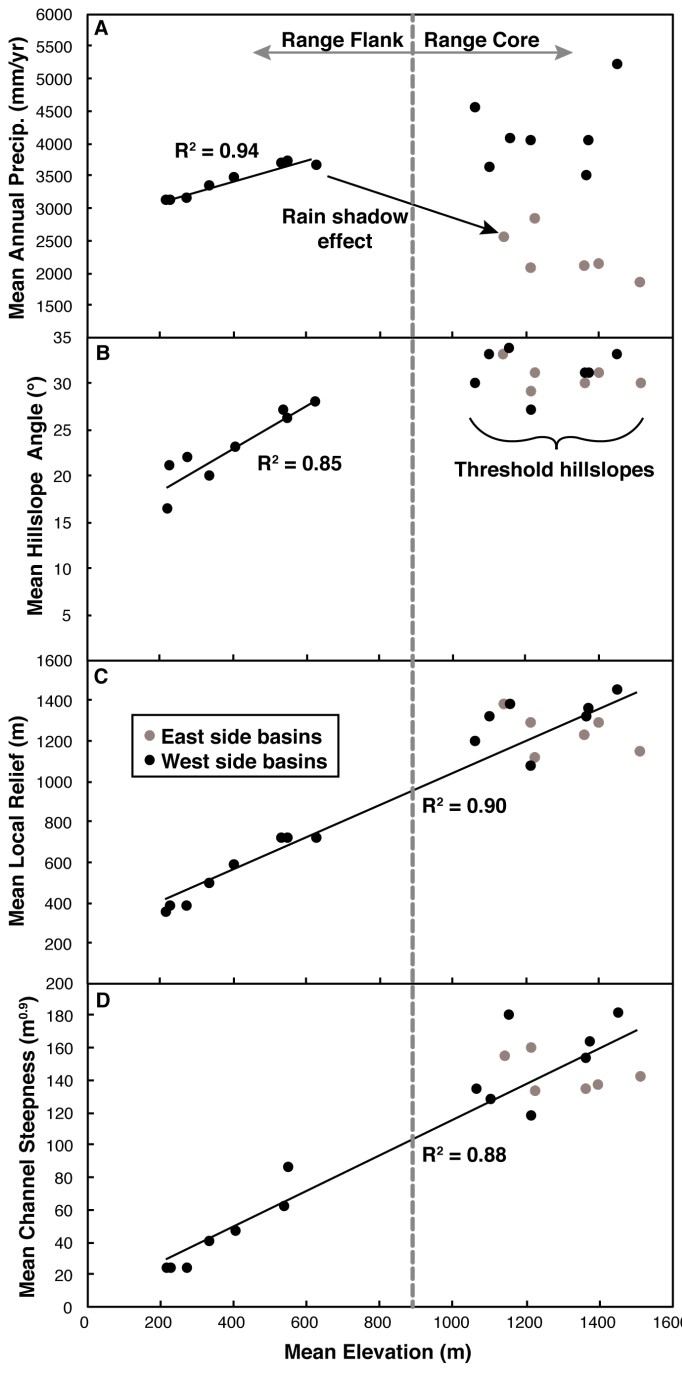


**Figure 3. Comparison of the mean basin elevation with other basin averaged metrics.** A) Mean annual precipitation. B) Hillslope angle. C) Local relief (using a 5-km diameter circle). D) Channel steepness.





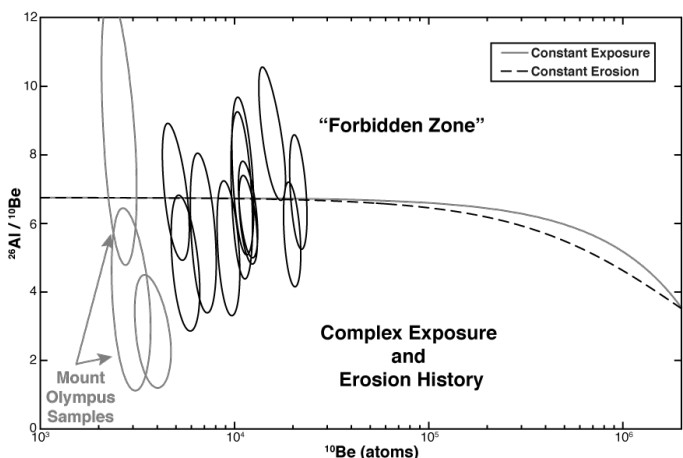

**Figure 4. Erosion island plot for new Olympic Mountain samples.** Each sample is represented by a 2σ error ellipse.
Dashed, grey ellipses mark samples with poor [10]Be measurements, see text for discussion.

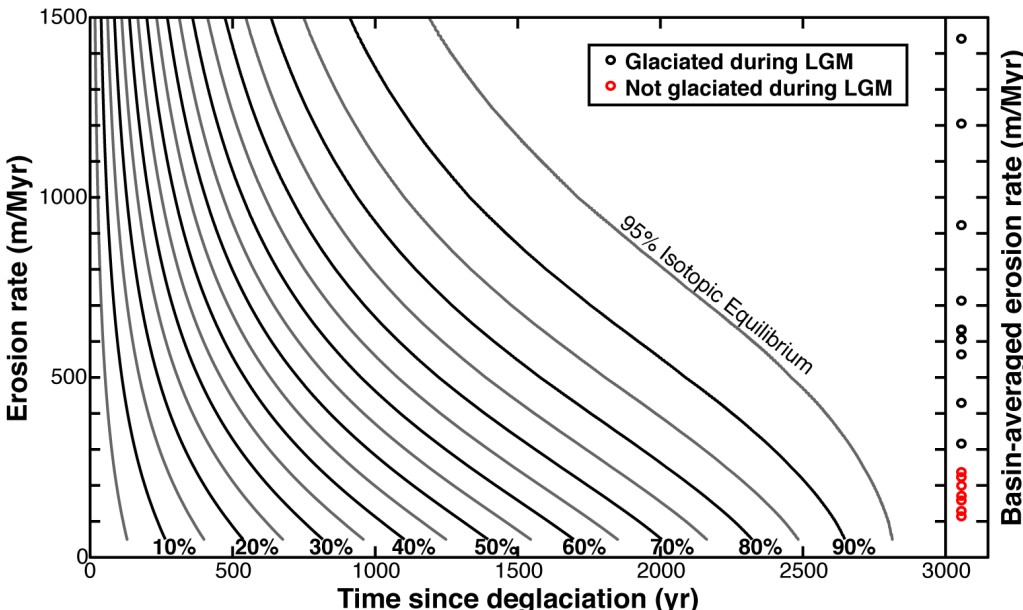

**Figure 5. Predicted evolution of landscapes toward isotopic evolution after deglaciation**. Black and grey lines denote
10% and 5% contour intervals, respectively. Basin-averaged erosion rates from the Olympic Mountains are shown on the
right. Basins marked in red were not glaciated during the Last Glacial Maximum (LGM).





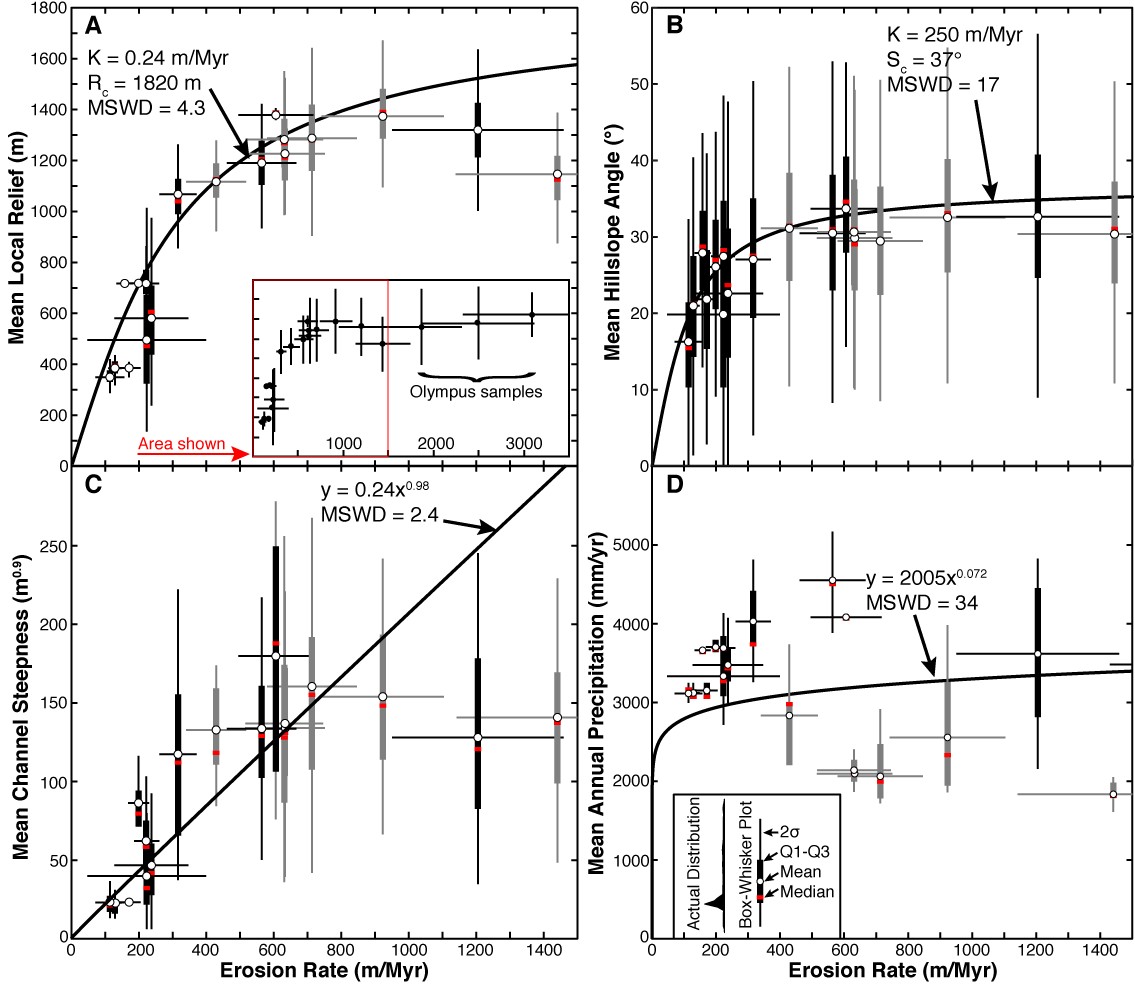

**Figure 6. Plot of erosion rates with basin-averaged metrics.** Due to the high degree of variation within a single basin, we have plotted basin metric data as box-and-whisker plots. Thin vertical bars denote 2 standard deviations. Thick vertical bars denote the central 50% of the data between the 1st and 3rd quartiles (Q1 and Q3). Red bars denote the medians and white circles denote means. Horizontal bars show $2\sigma$ confidence intervals on erosion rates. West and east side basins are shown in black and grey, respectively. A) Erosion rate versus mean local relief (5-km relief). Inset shows higher erosion rate samples not featured in other panels. B) Erosion rate versus mean hillslope angle. C) Erosion rate versus mean channel steepness. D) Erosion rate versus mean annual precipitation.



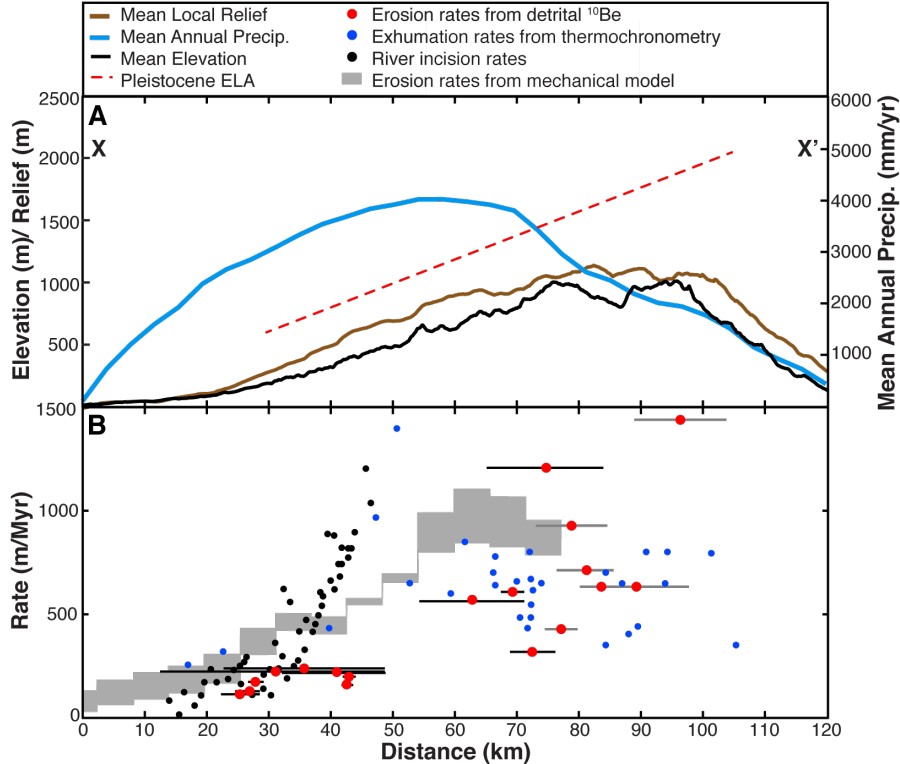

**Figure 7. Comparison between estimated erosion and relief across the Olympic Mountains.** A) Elevation and climate data across the Olympic Mountain range parallel to the direction of tectonic convergence (~54°). Maximum and mean elevations are shown in thin and thick black lines, respectively. Mean annual precipitation data (Daly et al., 1994) are shown in blue. Equilibrium line altitude (ELA) data (Porter, 1964) are represented by a red trend line. B) The blue circles show estimated rock uplift rates from apatite fission track data from Brandon et al. (1998). Black circles are rock uplift rate estimates from Pazzaglia and Brandon (2001) based on and river terrace incision. The grey envelope is the erosion pattern from Stolar et al. (2007) based on a coupled mechanical/landscape evolution model. Basin-average erosion rates in this study are shown in red circles with bars denoting the width of the basins. See Fig. 1A for cross section location.





**Table 1. Sample basin characteristics.** Mean equilibrium line altitiude (ELA) based on estimates from Porter (1964). 2σ = 2 standard deviations on the mean. Curves represent simplified histograms with normalized counts . See labels below each column for minimum and maximum bin values. Basins in italics are from Belmont et al. (2007).

| Sample Name | Latitude (°N) | Longitude (°W) | Area (km²) | Range Side | ELA (m) | Elevation (m) Mean | Elevation (m) 2σ | Elevation (m) Histogram | Mean Annual Prec. (mm/yr) Mean | Mean Annual Prec. (mm/yr) 2σ | Mean Annual Prec. (mm/yr) Histogram |
|---|---|---|---|---|---|---|---|---|---|---|---|
| WA1501 | 47.810972 | 123.44503 | 48.0 | East | 1673 | 1364 | 646 | | 2093 | 220 | |
| WA1502 | 47.948389 | 123.56092 | 66.8 | East | 1552 | 1215 | 786 | | 2058 | 626 | |
| WA1503 | 47.969639 | 123.59908 | 40.9 | East | 1431 | 1143 | 800 | | 2549 | 1139 | |
| WA1519 | 47.878306 | 123.70736 | 114.8 | West | 1413 | 1375 | 762 | | 4013 | 2275 | |
| WA1520 | 47.885139 | 123.75147 | 6.5 | West | 1255 | 1158 | 570 | | 4077 | 182 | |
| WA1522 | 47.976972 | 123.68797 | 19.2 | East | 1339 | 1230 | 520 | | 2831 | 911 | |
| WA1523 | 47.876735 | 123.69469 | 74.9 | West | 1449 | 1367 | 692 | | 3471 | 1710 | |
| WA1524 | 47.876161 | 123.69537 | 35.6 | West | 1342 | 1454 | 810 | | 5200 | 1478 | |
| WA1525 | 47.916688 | 123.24247 | 133.1 | East | 1811 | 1515 | 636 | | 1831 | 298 | |
| WA1526 | 47.67787 | 124.11701 | 126.9 | West | -- | 537 | 398 | | 3686 | 414 | |
| WA1527 | 47.62844 | 123.6316 | 115.4 | West | 1292 | 1064 | 652 | | 4544 | 761 | |
| WA1537 | 47.615017 | 123.47443 | 104.0 | West | 1438 | 1104 | 716 | | 3608 | 1678 | |
| WA1538 | 47.739067 | 123.17657 | 169.7 | East | 1734 | 1402 | 678 | | 2137 | 440 | |
| *WA1539* | *47.951718* | *123.81862* | *35.6* | West | 1270 | 1215 | 526 | | 4022 | 949 | |
| *U-EFMC* | *47.685616* | *124.23868* | *3.5* | West | -- | 275 | 162 | | 3150 | 179 | |
| *L-EFMC* | *47.653568* | *124.24006* | *13.4* | West | -- | 229 | 164 | | 3118 | 143 | |
| *U-WC* | *47.738694* | *124.04432* | *1.6* | West | -- | 629 | 234 | | 3659 | 109 | |
| *L-WC* | *47.728534* | *124.03657* | *4.3* | West | -- | 552 | 302 | | 3699 | 176 | |
| *DEN104* | *47.55637* | *124.28191* | *33.8* | West | -- | 220 | 158 | | 3111 | 143 | |
| *DEN106* | *47.644947* | *124.24263* | *281.2* | West | -- | 407 | 412 | | 3471 | 529 | |
| *DEN101* | *47.642348* | *124.23752* | *391.1* | West | -- | 335 | 422 | | 3328 | 644 | |
| | | | | | | 0 | 2450 | | 0 | 6000 | |



**Table 2. Basin metrics for erosion processes.** $2\sigma = 2$ standard deviations on the mean. Curves represent simplified histograms with normalized counts. See labels below each column for minimum and maximum bin values. Basins in italics are from Belmont et al. (2007). Values exclude data from ice coverd regions.

| Sample Name | Hillslope Angle (°) | | | Channel Steepness ($m^{0.9}$) | | | Local Relief (m) | | |
|---|---|---|---|---|---|---|---|---|---|
| | Mean | 2σ | Histogram | Mean | 2σ | Histogram | Mean | 2σ | Histogram |
| WA1501 | 30 | 20 | | 134 | 102 | | 1227 | 246 | |
| WA1502 | 29 | 20 | | 160 | 152 | | 1288 | 320 | |
| WA1503 | 33 | 22 | | 154 | 96 | | 1374 | 332 | |
| WA1519 | 31 | 24 | | 163 | 154 | | 1359 | 372 | |
| WA1520 | 34 | 19 | | 180 | 150 | | 1379 | 58 | |
| WA1522 | 31 | 20 | | 133 | 72 | | 1117 | 186 | |
| WA1523 | 31 | 24 | | 153 | 156 | | 1317 | 386 | |
| WA1524 | 33 | 26 | | 181 | 144 | | 1443 | 230 | |
| WA1525 | 30 | 20 | | 141 | 104 | | 1147 | 278 | |
| WA1526 | 27 | 20 | | 62 | 48 | | 717 | 174 | |
| WA1527 | 30 | 22 | | 134 | 96 | | 1190 | 242 | |
| WA1537 | 33 | 24 | | 128 | 106 | | 1320 | 292 | |
| WA1538 | 31 | 22 | | 137 | 124 | | 1282 | 334 | |
| WA1539 | 27 | 22 | | 117 | 106 | | 1067 | 224 | |
| *U-EFMC* | 22 | 18 | | 23.3 | 3.0 | | 386 | 28 | |
| *L-EFMC* | 21 | 18 | | 23 | 20 | | 385 | 72 | |
| *U-WC* | 28 | 16 | | N/A | N/A | N/A | 718 | 20 | |
| *L-WC* | 26 | 17 | | 86 | 28 | | 718 | 20 | |
| *DEN104* | 16 | 17 | | 23 | 17 | | 350 | 72 | |
| *DEN106* | 23 | 24 | | 47 | 50 | | 581 | 324 | |
| *DEN101* | 20 | 24 | | 40 | 48 | | 496 | 384 | |
| | | | 0        70 | | | 0        450 | | | 0        2100 |

735




**Table 3. Basin-averaged erosion rate sample data.** Integration time was calculated by dividing the e-folding depth of the production of cosmic nuclides via spallation (0.6 m) by the erosion rate. Italicized samples are from Belmont et al. (2007). Underlined sampes had [10]Be measurements less that 10 times the blank measurement.

| Sample Name | [10]Be (atoms/g) | [10]Be 2σ (atoms/g) | Be Erosion Rate (m/Myr) | Rate 2σ (m/Myr) | Integration Time (yr) | [26]Al (atoms/g) | [26]Al 2σ (atoms/g) | Al Erosion Rate (m/Myr) | Rate 2σ (m/Myr) | 26Al/10Be | 26Al/10Be 2σ |
|---|---|---|---|---|---|---|---|---|---|---|---|
| WA1501 | 11738 | 633 | 638 | 118 | 941 | 74391 | 6678 | 696 | 163 | 6.3 | 1.3 |
| WA1502 | 9324 | 527 | 718 | 134 | 836 | 48421 | 7135 | 959 | 321 | 5.2 | 1.6 |
| WA1503 | 6934 | 445 | 930 | 183 | 645 | 38878 | 6227 | 1152 | 414 | 5.6 | 1.9 |
| WA1519 | 2980 | 288 | 2511 | 618 | 239 | 10783 | 3104 | 4814 | 3104 | 3.6 | 2.2 |
| WA1520 | 10906 | 583 | 610 | 112 | 983 | 73333 | 10379 | 629 | 204 | 6.7 | 2.0 |
| WA1522 | 15665 | 1129 | 432 | 90 | 1389 | 132290 | 9907 | 353 | 75 | 8.4 | 1.8 |
| WA1523 | 3844 | 345 | 1881 | 442 | 319 | 10539 | 2435 | 4766 | 2429 | 2.7 | 1.4 |
| WA1524 | 2573 | 255 | 3117 | 782 | 193 | 21763 | 4289 | 2551 | 1113 | 8.5 | 3.7 |
| WA1525 | 5625 | 397 | 1451 | 301 | 414 | 26581 | 4208 | 2126 | 759 | 4.7 | 1.6 |
| WA1526 | 19765 | 845 | 224 | 37 | 2673 | 111196 | 11661 | 278 | 71 | 5.6 | 1.3 |
| WA1527 | 11048 | 596 | 564 | 104 | 1063 | 80390 | 9617 | 538 | 152 | 7.3 | 1.9 |
| WA1537 | 5010 | 376 | 1213 | 256 | 495 | 33985 | 3912 | 1244 | 341 | 6.8 | 1.9 |
| WA1538 | 11742 | 603 | 635 | 116 | 945 | 71021 | 6160 | 727 | 166 | 6.0 | 1.2 |
| WA1539 | 21267 | 915 | 318 | 55 | 1889 | 145691 | 13348 | 320 | 76 | 6.9 | 1.4 |
| *U-EFMC* | *21558* | *3018* | *171* | *34* | *3501* | -- | -- | -- | -- | -- | -- |
| *L-EFMC* | *27796* | *1668* | *129* | *20* | *4669* | -- | -- | -- | -- | -- | -- |
| *U-WC* | *29985* | *1799* | *158* | *25* | *3789* | -- | -- | -- | -- | -- | -- |
| *L-WC* | *22703* | *1362* | *199* | *31* | *3023* | -- | -- | -- | -- | -- | -- |
| *DEN-101* | *17407* | *11837* | *223* | *176* | *2685* | -- | -- | -- | -- | -- | -- |
| *DEN-104* | *31032* | *10551* | *114* | *43* | *5264* | -- | -- | -- | -- | -- | -- |
| *DEN-106* | *17150* | *7203* | *237* | *110* | *2528* | -- | -- | -- | -- | -- | -- |