# Peer review of "Tectonic controls of Holocene erosion in a glaciated orogen"

_Earth Surface Dynamics, 2018_

## Referee Comment (RC1) · Anonymous Referee #1 · 11 Apr 2018

General Comments:  The authors present results of topographic analysis and catchment-scale denudation rates determined using cosmogenic nuclides in the Olympic Mountains of Washington, USA. The goal of the analyses is to assess controls on the spatial patterns of denudation. The authors find that denudation rates scale with multiple metrics of topographic steepness and previously recognized spatial patterns of exhumation. Present day precipitation patterns and the extent of prior glaciation do not explain patterns denudation measured by cosmogenic nuclides.

The finding that denudation rates are not spatially correlated with precipitation, but instead are correlated with tectonic forcing is consistent with findings from a number of recent studies in other mountain ranges. Hence the work contributes to an emerging view on the role of climate in influencing erosion rates in tectonically-active landscapes

and is hence appropriate for publication in Earth Surface Dynamics. However, I have a number of comments that should be addressed in a revised manuscript below.

More generally, the manuscript begins (in the abstract) by indicating the role of topographic adjustment by glaciers in setting post-glacial erosion rates is unknown and that there are intense spatial variations in the glacial modification of topography. These statements are both correct, and in the Olympic Mountains there is evidence for spatial variation in topographic modification (e.g., Montgomery, 2002; Prasicek et al., 2014). However, the manuscript does not exploit the spatial variability in glacial modification to ask whether the degree of topographic modification and Holocene erosion rates scale with glacier size or whether Holocene erosion rates scale with the degree of glacial topographic modification. The manuscript stands on its own without addressing these questions, however, addressing these questions would increase the impact of the manuscript and help determine whether rock uplift alone drives the observed patterns in erosion or whether there is also an additional signature caused by glacier-induced increases in relief.

Comments: Line 21-24: It could easily be argued that buildup of topography, high relief, high erosion rates, etc., has also occurred after the onset of Cenozoic cooling and glaciation. Willett (1999) presents results of a modeling exercise, which does not directly support the claims regarding controls on topography and erosion prior to the Cenozoic. Hence it is not clear that these introductory sentences properly motivate the story that follows.

Line 24: 'between these characteristics' – please be explicit and write out what is meant by 'these'

Line 27: It would be useful to include a citation to a study or studies that document glacier fluctuations on response to climate change.

Line 47: It is not clear what is meant by 'efficiency' and also not clear what data/prior study support that statement.

Line 158: The description of why effective latitude and altitude values calculated for each catchment do not incorporate temporal variation in production rates needs to be re-visited or further explained. The time-variation in production is caused by temporal variation in earth's magnetic field. Given the size and elevation range of the catchments sampled, it is not clear (without a calculation to demonstrate it) that using effective altitude and latitude inputs would substantially distort predictions from a time-varying production model. Alternatively, simply state the reported values are based on a constant production rate model.

Line 194-196: The text explaining why statistics were not performed on subsets of the data is cumbersome, primarily because there first is not a justification for why the data would or could be divided into subsets.

Line 197: If the regressions account for uncertainty in both variables, then the regression technique should be reported; York or RMA or ?

Line 199: Given the MSWD statistic is little used in the geosciences outside of isochron geochronology, it would be useful to also report the correlation coefficient (R-squared) values.

Equations 3 and 4: It should be noted what values were used for K, Sc, and Rc.

Line 245: Here (and elsewhere, e.g., line 376) reference is made to the size of glaciers, but the manuscript does not report quantitative measures of glacier size, but instead refers to contours of ELA. Although they may be related, ELA is not the same as 'size'.

Line 271: A citation reporting the expected 26Al/10Be ratio is needed.

Line 343: An alternative explanation is that normalized channel steepness does not linearly track erosion.

Line 414: 'this study'; it is not clear if the phrase relates to the study cited in the previous sentence or to the present manuscript.

Line 406: The sentence is asking a question ('whether' appears twice), but then needs to end with a phrase that starts: 'depends on. . ."

Line 419-423: It is not clear that landscapes where glaciers are efficient agents of erosion are necessarily areas where glaciation reduces relief. It seems quite plausible that the excavation of glacial valleys/troughs could increase relief; indeed Montgomery (2002) reports that glacial valleys in the Olympic Mountains have 500 m more relief than fluvial valleys. Hence the text here needs reflect what is known empirically. Further, the following conclusion text (through line 423) is rather unsatisfactory, as these conclusions are not at all drawn from the findings presented in the manuscript. Such material could appear in the Discussion, albeit with less generalization, and a more robust discussion.

Line 424-425: References to mismatches between Holocene erosion and rock uplift seem better brought up in the Discussion; i.e., it is sufficient here to indicate there is a general agreement between erosion and rock uplift rates and to make conclusions based on that statement.

Tables: The topographic shielding factor should appear in one of the tables so that all data needed to re-produce the denudation rates are reported in the manuscript (see Frankel et al., 2010, EOS).

Editorial comments:

Line 68: there is a missing word 'pattern accreted materials'

Line 108-109 (and elsewhere): Several sentences begin with 'This'. Replacing 'This' with 'Equation 1. . .' or 'The value of 0.45. . .' would be clearer to the reader.

Line 144: lowercase 'v' in von.

Line 168: there is an extra word at the end of the sentence.

Line 220: it would be clearer to indicate that C and p are 'coefficients'

Line 311-312: this sentence is missing an ending

Line 410: there is extra text here; initials, first names

Figure 2e. The scale bar for erosion rates isn't very useful, as it is difficult to determine the rates for the catchments with yellow-green color.

Figure 3. The legend (east-, west-side basins) should appear in the top panel, because the text pointing out the rain shadow effect does not make sense without this information.

---

## Referee Comment (RC2) · G. Hilley (Referee) · 30 Apr 2018

Summary:

This contribution presents 14 new cosmogenically derived erosion rate measurements from the Olympic Mountains, Washington State, USA. These rates are compared to various climate measures, morphometrics, and exhumation / incision proxies to provide insight into the following questions: 1) Is there a discernible imprint of climate gradient on erosion rates measured by these cosmogenic inventories? 2) Is there a signal of disequilibrium conditions recorded by a discrepancy between erosion and exhumation measures over various time-scales?, 3) Do landscape morphometrics scale with erosion rates?, and 4) What is the relationship between measured erosion rates

and inferred long-term rock uplift rates? The authors generally find that variations in modern climate measures do not scale with measured erosion rates, but, at least at low erosion rates, measures such as local relief and mean channel steepness scale in some way with erosion rates. The authors find that there is a general correspondence between river incision rates, exhumation gauged by low-temperature thermochronology, and modern-day cosmogenically derived erosion rates. As such, even modern (millennial time-scale) erosion rates appear to track long-term exhumation (and perhaps rock-uplift) rates in the Olympics, and that glacial processes do not appear to disrupt landscape equilibrium to an extent that would produce a divergence between modern and long-term measures of erosion of the range.

Recommendation:

This paper presents interesting new data and analyses of an active, glaciated mountain belt, where large precipitation gradients and temporal changes in surface processes may be expected to leave some imprint on the erosion of the range. The authors finding that, despite these spatial and temporal variations, erosion rates measured over various time-scales are approximately constant, should be of interest to the readership of Earth Surface Dynamics. The study appears thoughtfully conceived and executed, and is written in a clear and concise manner that requires few grammatical changes. Thus, with consideration of the comments below, I would feel comfortable recommending acceptance pending MINOR to MODERATE REVISIONS. Below, I make some general suggestions, as well as some specific comments geared to individual lines in the text.

General Comments:

1) The authors have carried out a detrital 10-Be study that supposes that erosion rates in the catchments are everywhere equal. This is somewhat addressed in the text under the 5.1 section, last paragraph, where the authors discuss the impact of the introduction of dosed and shielded material into their samples. Yet, this does not address the

fact that the authors' approach assumes that each point in the basin is equally rep-
resented in the sample, as well as the fact that the calculated mean production rate
could be biased by increased contributions from different elevation ranges because of
the non-linear increase of production rate with elevation. I am not uncomfortable with
the authors' assumptions (in addition to the fact that quartz is uniformly distributed in
the sourced lithologies). But, given that some of these catchments have a good amount
of local relief and lithologic variability leading to heterogeneous quartz "fertility", some
discussion of this effect, and its potential impact might be appropriate to include in
section 5.1.

2) I found the correspondence between river incision, exhumation, model-derived ero-
sion rates, and 10-Be denudation rates compelling. One way in which these relation-
ships could be made more effective and illustrative would be to actually plot the quan-
tities versus one another, rather than distance (Figure 7). I think I understand why the
authors plotted these rates in the particular space they did, in that some of the primary
studies were carried out within areas that do not overlap with the cosmogenic samples
directly, but lie within similar tectonic positions when these data are projected onto a
cross section. My reading of the primary literature is that 1) the Clearwater (which I
think are the black dots) is located outside of the sampled area shown in Figure 2E,
and so must be projected into the sampled basins to be used in this study. 2) The AFT
ages are from throughout the range, and so there is probably a good path forward for
interpolating these across the sampled basins to calculate point-by-point estimates of
exhumation rate, and to use these to quantify basin-averaged exhumation rates within
each sampled watershed. 3) Drew Stolar's modeling study is a profile model, which is
fine. But, it is tuned to a specific mean erosion rate that I think was chosen with the
AFT exhumation rates in mind. Thus, it is not particularly surprising that the magni-
tudes match up with what is observed, since the AFT exhumation rates roughly align
with the 10-Be rates.

This is all to say that the most robust and comparable of these datasets appear to be the

AFT- and 10-Be-derived rates. To use the other measures, it seems like some sort of projection has to be made, which introduces its own set of assumptions. Please correct me if I am wrong about this. This being the case, it might be interesting to carry out a direct comparison of the most comparable of the datasets, meaning the (interpolated) AFT-derived exhumation rates for each watershed that was sampled for 10-Be. One would then have a direct comparison that could be used to carry out a formal analysis to reveal the strength of correspondence (through regression analysis), the presence / absence of systematic discrepancies between datasets that could reveal more insight into where correspondences are best versus where they might break down to some degree. I feel that a map of the difference between exhumation and erosion rates in the spirit of Figure 2E might be revealing.

Specific Comments:

Lines 115-120: For documentation, it would probably be good to have some description of how you calculated ks. There are a few different solution methods available, and a few different software packages to do this. Quick mention of these would be good for completeness of documentation.

Lines 290-300: The relationship with channel steepness looks good between $0 \leq e \leq 600$ m/Myr. At higher erosion rates, it does not do so well, even for the samples with reasonable 26Al/10Be. Also, you might consider adjusting your concavity to 0.4, so that you can use the multiplier of the power law to calculate K and compare this to the range for your lithologies via Stock and Montgomery (1999). This will help provide some additional validation.

Lines 310-315: As per general comment 1 above, I don't think that spatial variations in erosion rates are covered here, meaning that one could have a 6.75 ratio (i.e., no complex exposure history) but your sample might not be representative of all sediment in the basin. Also, I assume that the "deep" landslide mechanism would be one in which dosed material was buried for a long time (a fraction of the 1/2 life of 26-Al)

and then reworked into transported sediment. But, I think a "deep" bedrock landslide should still have the 6.75 ratio, since it did not get dosed and re-buried.

Thanks for the opportunity to review this work. I really enjoyed the paper and look forward to its formal "release" online in Earth Surface Dynamics.

George Hilley.

---

## Author Comment (AC1) · 29 May 2018

**Reviewer #1**
(Responses in blue)
General Comments:
The authors present results of topographic analysis and catchment-scale denudation rates determined using cosmogenic nuclides in the Olympic Mountains of Washington, USA. The goal of the analyses is to assess controls on the spatial patterns of denudation. The authors find that denudation rates scale with multiple metrics of topographic steepness and previously recognized spatial patterns of exhumation. Present day precipitation patterns and the extent of prior glaciation do not explain patterns denudation measured by cosmogenic nuclides.

The finding that denudation rates are not spatially correlated with precipitation, but instead are correlated with tectonic forcing is consistent with findings from a number of recent studies in other mountain ranges. Hence the work contributes to an emerging view on the role of climate in influencing erosion rates in tectonically-active landscapes and is hence appropriate for publication in Earth Surface Dynamics. However, I have a number of comments that should be addressed in a revised manuscript below.

More generally, the manuscript begins (in the abstract) by indicating the role of topographic adjustment by glaciers in setting post-glacial erosion rates is unknown and that there are intense spatial variations in the glacial modification of topography. These statements are both correct, and in the Olympic Mountains there is evidence for spatial variation in topographic modification (e.g., Montgomery, 2002; Prasicek et al., 2014).

However, the manuscript does not exploit the spatial variability in glacial modification to ask whether the degree of topographic modification and Holocene erosion rates scale with glacier size or whether Holocene erosion rates scale with the degree of glacial topographic modification. The manuscript stands on its own without addressing these questions, however, addressing these questions would increase the impact of the manuscript and help determine whether rock uplift alone drives the observed patterns in erosion or whether there is also an additional signature caused by glacier-induced increases in relief.

We agree that this is an important and interesting topic; however, it goes beyond the scope of this manuscript. Such an endeavor would require a new study that focused on quantifying ice volumes and or ELA histories throughout the range. As it stands, evidence of non-eroded moraines is limited to the western side of the range, and estimates of glacier size can only be deduced using the elevations of cirque basins and impeding ice sheet lobes on the eastside of the range, all estimates and observations that we have utilized in this manuscript, and expanded upon in further suggestions below.

Comments:
Line 21-24: It could easily be argued that buildup of topography, high relief, high erosion rates, etc., has also occurred after the onset of Cenozoic cooling and glaciation. Willett (1999) presents results of a modeling exercise, which does not directly support the claims regarding controls on topography and erosion prior to the Cenozoic. Hence it is not clear that these introductory sentences properly motivate the story that follows.

We agree with the reviewer on this topic. However, the point we are making first is that there must have been Cenozoic mountain ranges present in order to form large areas of generally cold landscapes were snow/ice could accumulate to form alpine glaciers. Without this precondition, alpine glaciers would not have formed. After these sentences, we then discuss the idea that glaciers changed topography and erosion thereafter. To communicate this point better we have modified the opening of the Introduction (4$^{th}$ sentence) to read – "This increase

in cooler, higher elevation landscapes created the necessary conditions for alpine glaciers to form in the Late Cenozoic."

Line 24: 'between these characteristics' – please be explicit and write out what is meant by 'these'

Corrected: This sentence now reads – "Because of the covariation between climate, topography and rock uplift, erosion rates in fluvially dominated orogens have been shown to correlate with climatic and topographic metrics such as precipitation rate, relief, hillslope angle, and channel steepness via linear, non-linear and threshold relationships…"

Line 27: It would be useful to include a citation to a study or studies that document glacier fluctuations on response to climate change.

We have decided to remove this superfluous sentence to make the Introduction more to the point.

Line 47: It is not clear what is meant by 'efficiency' and also not clear what data/prior study support that statement.

Corrected: We have changed this sentence to read – "Here we address this uncertainty and test whether Plio-Pleistocene glaciers have masked long-lived patterns of rock uplift as recorded by millennial-scale erosion rate estimates and modern topography."

Line 158: The description of why effective latitude and altitude values calculated for each catchment do not incorporate temporal variation in production rates needs to be re-visited or further explained. The time-variation in production is caused by temporal variation in earth's magnetic field. Given the size and elevation range of the catchments sampled, it is not clear (without a calculation to demonstrate it) that using effective altitude and latitude inputs would substantially distort predictions from a time varying production model. Alternatively, simply state the reported values are based on a constant production rate model.

Corrected: This sentence now reads – "We report erosion rates from the CRONUS calculator from the constant production rates determined by the constant production rate models of Lal (1991) and Stone (2000). To enable comparison between new and previous measurements, we recalculated erosion rates from 7 sand samples within the Olympic Mountains previously reported by Belmont et al. (2007)."

Line 194-196: The text explaining why statistics were not performed on subsets of the data is cumbersome, primarily because there first is not a justification for why the data would or could be divided into subsets.

Corrected: We agree these statements are not necessary. These two sentences have been removed.

Line 197: If the regressions account for uncertainty in both variables, then the regression technique should be reported; York or RMA or ?

Corrected: The first two sentences of this section now read – "We performed non-linear, least-square regressions on our new and existing erosion rate data. To provide a better sense of the distribution of topographic metrics within a basin, we provide box-and-whisker plots within our bivariate plots, though our regressions discussed in the following sections are based on

mean statistics. We included the uncertainties in both variables by using a Mote Carlo sampling protocol."

Line 199: Given the MSWD statistic is little used in the geosciences outside of isochron geochronology, it would be useful to also report the correlation coefficient (R-squared) values.

We agree that the MSWD is not often used in the geosciences, and find this unfortunate. However, standard $R^2$ values are not appropriate for these regressions as they are not linear, and therefore, $R^2$ values can exceed 1, thus making it a poor indicator of goodness of fit. As the MSWD is a standard data analysis tool that is well documented in text books, and appropriate for the data sets with the properties such as ours we have chosen to use it.

Equations 3 and 4: It should be noted what values were used for K, Sc, and Rc.

There are no set values for K, Sc, and Rc. These values are found through the regression techniques and reported in the following sections.

Line 245: Here (and elsewhere, e.g., line 376) reference is made to the size of glaciers, but the manuscript does not report quantitative measures of glacier size, but instead refers to contours of ELA. Although they may be related, ELA is not the same as 'size'.

Corrected: We have clarified this topic by adding the following text to the background section – "Alpine glaciers were likely active in every valley of the Olympics (Porter, 1964); however, the size of the glaciers was highly variable, as the east flowing glaciers would have been limited to the rugged core of the range by the Cordilleran Ice Sheet (see glacial deposits Fig. 1B). This suggests that the west flowing glaciers may have been nearly twice as long as those flowing east."

Line 271: A citation reporting the expected 26Al/10Be ratio is needed.

Corrected: A citation to Balco et al 2008 added.

Line 343: An alternative explanation is that normalized channel steepness does not linearly track erosion.

We agree with this observation, and discuss it in a previous section (3.4) when describing our non-linear regression technique.

Line 414: 'this study'; it is not clear if the phrase relates to the study cited in the previous sentence or to the present manuscript.

Corrected: "this study" changed to "these numerical models" for clarification that the reference was to the previous study.

Line 406: The sentence is asking a question ('whether' appears twice), but then needs to end with a phrase that starts: 'depends on. . ."

Corrected: We have restructured this sentence to be much clearer – "The balance between post-glacial erosion and longer-lived rock uplift depends on whether post-glacial climate conditions (e.g. increase or decrease in precipitation), or topographic perturbations (e.g. hillslope steeping or channel shallowing) have changed the activity of extant surface processes."

Line 419-423: It is not clear that landscapes where glaciers are efficient agents of erosion are necessarily areas where glaciation reduces relief. It seems quite plausible that the excavation of glacial valleys/troughs could increase relief; indeed Montgomery (2002) reports that glacial valleys in the Olympic Mountains have 500 m more relief than fluvial valleys. Hence the text here needs reflect what is known empirically.

Indeed, glacial incision can increase or decrease relief. Whether an increase or decrease is detected is a matter of scale of observation. There are a few studies which have shown this to occur. We have added references to a well-known study (Whipple et al., 2009) and our recent paper in the Olympic Mountains (Adams and Ehlers, 2017) to provide context.

Further, the following conclusion text (through line 423) is rather unsatisfactory, as these conclusions are not at all drawn from the findings presented in the manuscript. Such material could appear in the Discussion, albeit with less generalization, and a more robust discussion.

Corrected: We have moved most of this material to the Discussion section and reshaped the Conclusion.

Line 424-425: References to mismatches between Holocene erosion and rock uplift seem better brought up in the Discussion; i.e., it is sufficient here to indicate there is a general agreement between erosion and rock uplift rates and to make conclusions based on that statement.

Corrected: We have moved most of this material to the Discussion section and reshaped the Conclusion.

Tables: The topographic shielding factor should appear in one of the tables so that all data needed to re-produce the denudation rates are reported in the manuscript (see Frankel et al., 2010, EOS).

We have opted to include these data in the supplement for the sake of space in the main manuscript.

Editorial comments:
Line 68: there is a missing word 'pattern accreted materials'

Corrected: This now reads – "pattern of accreted materials"

Line 108-109 (and elsewhere): Several sentences begin with 'This'. Replacing 'This' with 'Equation 1. . .' or 'The value of 0.45. . .' would be clearer to the reader.

Corrected: Sentences now read – "Eq. 1 normalizes slope values, for the concavity of the channel. For our calculations, we use $\theta = 0.45$. A $\theta$ value of 0.45 has been shown to describe the concavity of fluvial systems in the Olympic Mountains (Adams and Ehlers, 2017)."
Some other instances throughout the text have also bee changed.

Line 144: lowercase 'v' in von.

Corrected.

Line 168: there is an extra word at the end of the sentence.

Corrected. The extra "the" has been removed.

Line 220: it would be clearer to indicate that C and p are 'coefficients'

Corrected.

Line 311-312: this sentence is missing an ending

Corrected. Sentence now reads – "Recent studies suggest that our samples should have natural $^{26}Al/^{10}Be$ ratios of ~6.75 (Balco et al., 2008), a value that is very close to most of our measured ratios (Figure 4)."

Line 410: there is extra text here; initials, first names

Corrected.

Figure 2e. The scale bar for erosion rates isn't very useful, as it is difficult to determine the rates for the catchments with yellow-green color.

Corrected. Scale now runs only from red to blue.

Figure 3. The legend (east-, west-side basins) should appear in the top panel, because the text pointing out the rain shadow effect does not make sense without this information.

Corrected.

**Reviewer #2 (George Hilley)**
(Responses in blue)
Summary:
This contribution presents 14 new cosmogenically derived erosion rate measurements from the Olympic Mountains, Washington State, USA. These rates are compared to various climate measures, morphometrics, and exhumation / incision proxies to provide insight into the following questions: 1) Is there a discernible imprint of climate gradient on erosion rates measured by these cosmogenic inventories? 2) Is there a signal of disequilibrium conditions recorded by a discrepancy between erosion and exhumation measures over various time-scales?, 3) Do landscape morphometrics scale with erosion rates?, and 4) What is the relationship between measured erosion rates and inferred long-term rock uplift rates? The authors generally find that variations in modern climate measures do not scale with measured erosion rates, but, at least at low erosion rates, measures such as local relief and mean channel steepness scale in some way with erosion rates. The authors find that there is a general correspondence between river incision rates, exhumation gauged by low-temperature thermochronology, and modern-day cosmogenically derived erosion rates. As such, even modern (millennial time-scale) erosion rates appear to track long-term exhumation (and perhaps rock-uplift) rates in the Olympics, and that glacial processes do not appear to disrupt landscape equilibrium to an extent that would produce a divergence between modern and long-term measures of erosion of the range.

Recommendation:
This paper presents interesting new data and analyses of an active, glaciated mountain belt, where large precipitation gradients and temporal changes in surface processes may be

expected to leave some imprint on the erosion of the range. The authors finding that, despite these spatial and temporal variations, erosion rates measured over various time-scales are approximately constant, should be of interest to the readership of Earth Surface Dynamics. The study appears thoughtfully conceived and executed, and is written in a clear and concise manner that requires few grammatical changes. Thus, with consideration of the comments below, I would feel comfortable recommending acceptance pending MINOR to MODERATE REVISIONS. Below, I make some general suggestions, as well as some specific comments geared to individual lines in the text.

General Comments:

1) The authors have carried out a detrital 10-Be study that supposes that erosion rates in the catchments are everywhere equal. This is somewhat addressed in the text under the 5.1 section, last paragraph, where the authors discuss the impact of the introduction of dosed and shielded material into their samples. Yet, this does not address the fact that the authors' approach assumes that each point in the basin is equally represented in the sample, as well as the fact that the calculated mean production rate could be biased by increased contributions from different elevation ranges because of the non-linear increase of production rate with elevation. I am not uncomfortable with the authors' assumptions (in addition to the fact that quartz is uniformly distributed in the sourced lithologies). But, given that some of these catchments have a good amount of local relief and lithologic variability leading to heterogeneous quartz "fertility", some discussion of this effect, and its potential impact might be appropriate to include in section 5.1.

Corrected. We have added the following discussion in section 5.1 – "Like a previous study (Belmont et al, 2007) we have assumed that there is no risk to erosion rate calculations due to quartz infertility or proportional quartz sourcing from all parts of our basins in the Olympic Mountains. While there are some quartz-free lithologies in the range, these rocks are a minor occurrence the in Olympic Subduction Complex, and we have avoided sampling the Coast Range Terrane completely. In locations where nested catchments are found, erosion rates are with error of each other, suggesting a proportional scouring of quartz from all parts of even the largest sampled basin (compare WA1526, DEN101, and DEN106)."

2) I found the correspondence between river incision, exhumation, model-derived erosion rates, and 10-Be denudation rates compelling. One way in which these relationships could be made more effective and illustrative would be to actually plot the quantities versus one another, rather than distance (Figure 7). I think I understand why the authors plotted these rates in the particular space they did, in that some of the primary studies were carried out within areas that do not overlap with the cosmogenic samples directly, but lie within similar tectonic positions when these data are projected onto a cross section. My reading of the primary literature is that 1) the Clearwater (which I think are the black dots) is located outside of the sampled area shown in Figure 2E, and so must be projected into the sampled basins to be used in this study.

The river incision data (black dots) are within the sampled basins (most the CRN data from the previous study. As such there is substantial overlap between the datasets.

2) The AFT ages are from throughout the range, and so there is probably a good path forward for interpolating these across the sampled basins to calculate point-by-point estimates of exhumation rate, and to use these to quantify basin-averaged exhumation rates within each sampled watershed.

Indeed, we have considered a method very similar to what is being described here. However, we found the method of interpolating/extrapolating to be unsatisfactory as a means of direct comparison with our data for the following reasons. 1) The integrating spatial scales of erosion are not clear for AFT data. What portion of the landscape surrounding a bedrock sample could be exhuming at the rate recorded using AFT data? 2) An interpolating/extrapolating method will create a smooth function between AFT points while the erosion field from CRN data will be discontinuous. 3) The density of AFT data throughout the range is variable, and only 4 of our new samples in the core of the range with acceptable erosion rates contain an AFT sample. Taken together, it would be difficult to constrain the uncertainty introduced into the regression by using interpolated/extrapolated data.

3) Drew Stolar's modeling study is a profile model, which is fine. But, it is tuned to a specific mean erosion rate that I think was chosen with the AFT exhumation rates in mind. Thus, it is not particularly surprising that the magnitudes match up with what is observed, since the AFT exhumation rates roughly align with the 10-Be rates.

This is a fair point. We have opted to remove these modeling results to make the means of comparison more straightforward.

This is all to say that the most robust and comparable of these datasets appear to be the AFT- and 10-Be-derived rates. To use the other measures, it seems like some sort of projection has to be made, which introduces its own set of assumptions. Please correct me if I am wrong about this. This being the case, it might be interesting to carry out a direct comparison of the most comparable of the datasets, meaning the (interpolated) AFT-derived exhumation rates for each watershed that was sampled for 10-Be. One would then have a direct comparison that could be used to carry out a formal analysis to reveal the strength of correspondence (through regression analysis), the presence / absence of systematic discrepancies between datasets that could reveal more insight into where correspondences are best versus where they might break down to some degree. I feel that a map of the difference between exhumation and erosion rates in the spirit of Figure 2E might be revealing.

Please see comments above for more on this topic. It is true that compressing the data from the Olympic Mountains to a single profile requires some assumptions; however, we believe that this technique introduces less bias, and is less likely to over interpret the data available, than a direct 1:1 comparison of AFT and CRN rates.

Specific Comments:
Lines 115-120: For documentation, it would probably be good to have some description of how you calculated ks. There are a few different solution methods available, and a few different software packages to do this. Quick mention of these would be good for completeness of documentation.

Corrected: We have added a new sentence in this section that reads: "We used the Profiler tool (Wobus et al., 2006) to extract and analyze our river channels, and calculated steepness values over 0.5 km reaches."

Lines 290-300: The relationship with channel steepness looks good between $0 \leq e \leq 600$ m/Myr. At higher erosion rates, it does not do so well, even for the samples with reasonable 26Al/10Be.

This appears to be more of an observation (which we completely agree with) than a concern that needs addressing. We have made the same observation and discuss how this pattern may be important in glaciated landscape where topographic relief has been limited by glaciers.

Also, you might consider adjusting your concavity to 0.4, so that you can use the multiplier of the power law to calculate K and compare this to the range for your lithologies via Stock and Montgomery (1999). This will help provide some additional validation.

This is an interesting idea; however, the data and analysis presented here does not allow for the precise vetting of a specific incision law. The power law in Eq 5 may not be recording the stream power law in Stock and Montgomery because our new data may not be in topographic steady-state. To compare the power law coefficient from our data with that of Stock and Montgomery would be an over interpretation.

Lines 310-315: As per general comment 1 above, I don't think that spatial variations in erosion rates are covered here, meaning that one could have a 6.75 ratio (i.e., no complex exposure history) but your sample might not be representative of all sediment in the basin. Also, I assume that the "deep" landslide mechanism would be one in which dosed material was buried for a long time (a fraction of the 1/2 life of 26-Al) and then reworked into transported sediment. But, I think a "deep" bedrock landslide should still have the 6.75 ratio, since it did not get dosed and re-buried.

This topic is covered in detail above after General Comment 1. In addition, we have removed "deep landslide" reference from this sentence as it was incorrectly added there.

[revised manuscript text omitted]